# CSN5A Subunit of COP9 Signalosome Is Required for Resetting Transcriptional Stress Memory after Recurrent Heat Stress in *Arabidopsis*

**DOI:** 10.3390/biom11050668

**Published:** 2021-04-30

**Authors:** Amit Kumar Singh, Shanmuhapreya Dhanapal, Alin Finkelshtein, Daniel A. Chamovitz

**Affiliations:** 1Jacob Blaustein Institutes for Desert Research, Ben-Gurion University of the Negev, Midreshet Ben-Gurion 8499000, Israel; shanmuhapreya@gmail.com (S.D.); dchamovitz@bgu.ac.il (D.A.C.); 2School of Plant Sciences and Food Security, Tel Aviv University, Tel Aviv 6997801, Israel; alinfin@gmail.com

**Keywords:** COP9 signalosome, hypomorphic mutant, transcriptional stress memory, histone methylation, recurrent heat stress

## Abstract

In nature, plants are exposed to several environmental stresses that can be continuous or recurring. Continuous stress can be lethal, but stress after priming can increase the tolerance of a plant to better prepare for future stresses. Reports have suggested that transcription factors are involved in stress memory after recurrent stress; however, less is known about the factors that regulate the resetting of stress memory. Here, we uncovered a role for Constitutive Photomorphogenesis 5A (CSN5A) in the regulation of stress memory for resetting transcriptional memory genes (*APX2* and *HSP22*) and H3K4me3 following recurrent heat stress. Furthermore, CSN5A is also required for the deposition of H3K4me3 following recurrent heat stress. Thus, CSN5A plays an important role in the regulation of histone methylation and transcriptional stress memory after recurrent heat stress.

## 1. Introduction

The COP9 (constitutive photomorphogenesis 9) Signalosome (CSN) is an evolutionarily conserved protein complex consisting of eight subunits (CSN1 to CSN8) in higher eukaryotes such as plants, mammals, and insects [1]. It was first identified as a repressor of light-dependent growth in plants [2,3]. The regulation of protein degradation through the removal of ubiquitin-like protein Nedd8 (deneddylation) from the Cullin–RING E3 ligase (CRL) is the most studied biochemical function of CSN. The CSN is a metalloprotease, and the catalytic activity for deneddylation is present in the JAMM (JAB1/MPN/Mov34 metaloenzyme) motif of the CSN5 subunit [4]. CSN regulates several hormonal signals through its deneddylase activity [5]. However, deneddylation is not the only function of CSN [6,7,8,9,10]. Through these functions, CSN is involved in maintaining the stability of transcription factors, and thus, downstream transcription as well [11,12]. CSN subunits localize to chromatin, suggesting that the CSN regulates transcription directly [6,13]. In plants, the CSN is involved in various developmental processes including photomorphogenesis [3], floral development [14], cell-cycle regulation [11], and defense response [15], which cannot be exclusively explained by the deneddylation on CRLs because mutants of a related ubiquitin pathway do not phenocopy CSN mutants. Using a *csn5a-1* hypomorphic mutant, it was shown that CSN-dependent gene expression was partially regulated by CSN-dependent DNA methylation [9], which suggests that CSN is involved in epigenetic regulation.

Plants are sessile in nature and thus must withstand stressful environmental conditions for survival and reproduction. Plants remember past stressful experiences to be better prepared for subsequent incidences [16,17]. Stress memory can be regulated at multiple levels, from the metabolite to chromatin levels. Stress memory related to gene expression is potentially regulated by chromatin structure. In plants, H3K4 trimethylation correlates with active transcription [18] and RNA polymerase II elongation [19]. However, H3K4me2 is correlated with the 5′ region of gene but not with active transcription [18]. Using heat stress memory-related genes HSP22 and ascorbate peroxidase 2 (APX2), it was found that the HSFA2 transcription factor is important for the transcriptional response to recurring heat stress. Furthermore, H3K4 di and trimethylations were enriched in the memory genes [20]. Heat stress memory genes are classified by a higher expression that can persist for a few days following recurrent heat stress [21]. In another study, H3K4me3 was not correlated with a transcriptional response after priming treatment at the individual gene level [17], indicating that H3K4me3 does not always correlate with active transcription. The study of stress memory is an emerging field where the roles of a few components such as BRUSHY1 (BRU1)/TONSOKU/MGOUN3, FORGETTER1, and HSFA2 are known to maintain stress-induced chromatin memory for a few days [22]. However, it is not known why the memory is maintained only for a few days and how it is reset thereafter.

In a previous study, we found that the CSN5A subunit of CSN buffers the response to recurrent heat stress by maintaining auxin signaling [23]. In the *csn5a-1* hypomorphic mutant, we found that auxin signaling maintained a higher level even after 3 d of recurring heat stress whereas it reverted to the baseline level in the wild type, indicating that *csn5a-1* retains the memory of stress. Here, we hypothesize that gene expression and histone methylation after heat stress are regulated by CSN and conduct an experiment to prove our hypothesis.

## 2. Materials and Methods

### 2.1. Plant Material and Growth Conditions

The *Arabidopsis* line with a Columbia-0 (Col-0) background was used in this work. The transgenic line of *csn5a-1* was illustrated previously [24]. In brief, *csn5a-1* showed reduced growth with lower later root, deneddylation activity, and auxin response. Sterile seeds were sown on 1× Murashige and Skoog salt (MS) medium, 0.8% agar, 1% sucrose, 0.05% MES, pH 5.7. After two days of cold stratification (4 °C in dark), the plates were transferred to the growth chamber at 21 °C under long day conditions (16 h white light at 100 µmol m^2^s^−1^ and 8 h darkness) at 70% relative humidity. Ten days after sowing (DAS), the seedlings were transferred to the soil and stress treatment was given at 21 DAS.

### 2.2. Heat Treatments 

The plants were treated with 44 °C for 2 hours a day, starting at 11:00, for 7 days. A schematic representation of the experimental design is shown in Figure 1. The relative humidity of the chamber was maintained at 55–70% during heat treatment.

### 2.3. Chromatin Immunoprecipitation 

Chromatin Immunoprecipitation (ChIP) was performed following the method in Saleh et al., 2008 [25], with little modification. In short, a control using approximately 4 g of 28- and 31-DAS-old control and heat-treated seedlings were taken 3 h and 3 d after the heat treatment, and they were cross linked with 1% formaldehyde by vacuum infiltration twice for 10 min on ice. The samples were frozen in liquid nitrogen and stored at −80 °C until further processing. The nuclei were extracted, and the chromatins were sonified using a precooled Biorupter (10 cycles, 30 s on/off) (Diagenode, Seraing, Belgium). Equal amounts of chromatin from the same preparation were immunoprecipitated with antibodies (anti-H3, ab1791; anti-H3K4me2, ab11946; and anti-H3K4me3, ab8580; all from Abcam) at 4 °C overnight. Chromatin and protein immunoprecipitate were incubated further for 2 h with protein A/G magnetic beads (Thermo Fisher Scientific) at 4 °C. The magnetic beads were washed with low salt, high salt, LiCl, and TE buffer. Reverse cross-linking was performed as described in the protocol in Reference [25]. DNA was extracted using phenol/chloroform/isoamyl alcohol (25:24:1) and precipitated with 100% ethanol, sodium acetate, and glycogen (Thermo Fisher Scientific). The precipitated DNA was dissolved in 50 µl of TE and stored at −80 °C until qPCR was performed. A list of primers used for ChIP qPCR is given in Appendix A. Amplification values were normalized to H3 and the respective non-heat stress sample.

### 2.4. qRT-PCR Analysis 

RNA was extracted from 28- and 31-DAS-old control and heat-treated seedlings 3 h and 3 d after the heat treatment using the TriReagent (MRC) method. cDNA was synthesized using qScript cDNA Synthetic Kit (Quantabio, Beverly, MA, USA), and qRT-PCR was performed using PerfectCT SYBR Green FastMix (Quantabio, Beverly, MA, USA). *ACTIN* was used as normalization gene. The experiments were performed in three biological replicates with three technical replications for each biological sample. A list of primers and genes used for qRT-PCR is given in Appendix A. For statistical analysis, unpaired Student’s t test was used.

## 3. Results

### 3.1. CSN5A Regulates the Expression of Heat Stress Memory Genes APX2 and HSP22 But Not HSP70 after Recurrent Heat Stress

To study whether the expression profiles of the heat stress memory-related and non-memory genes depends on CSN5A following recurrent heat stress (2 h, 44 °C, 7 d), we investigated two memory genes (*APX2* and *HSP22*) and one non-memory gene (*HSP70*) [20]. We observed a 2.73-fold increase in *APX2* in Col-0 3 h after the heat stress regimen compared to the control Col-0. However, *csn5a-1* showed a 9.7-fold change in *APX2* expression 3 h following heat treatment, which was significantly higher than that in Col-0. Three days following the heat regimen, *APX2* expression in Col-0 returned to the baseline while *csn5a-1* remained elevated 2.3-fold (Figure 2A). *HSP22* expression increased drastically in Col-0 and *csn5a-1* 3 h following the heat regimen; however, the fold change of *HSP22* in *csn5a-1* (2108) was two times higher than that in Col-0 (1023). Similar to *APX2*, the *HSP22* levels also reached to the baseline after 3 d of heat treatment in Col-0 whereas *csn5a-1* still showed a 4.96-fold higher expression of *HSP22* (Figure 2B). Interestingly, Col-0 showed a 2.84-fold higher expression of *HSP70* 3 h following heat treatment, but no elevation was found in the *csn5a-1* mutant. Moreover, *HSP70* expression did not reach the baseline 3 d after heat stress in Col-0; instead, it increased to 3.5-fold. Thus, the result suggests that CSN5A regulates the expression of heat stress memory genes (*APX2* and *HSP22*) but not *HSP70* by controlling the expression of memory genes 3 h following heat stress. Furthermore, CSN5A is required to return the levels of memory genes to the baseline 3 d after heat treatment.

### 3.2. Expression of Heat Stress Memory Genes Is Not Associated with H3K4 Methylation 

To study the association between heat stress memory genes and chromatin modification, we investigated H3K4me3 and H3K4me2 at the *APX2* and *HSP22* loci as representatives of memory genes and at the *HSP70* loci as a non-memory gene. For all three genes, we examined one genic region near the 5′ end of the gene and one flanking region at least 2.5 kb away in the intergenic region. We observed 2.82- and 4.9-fold enrichments in H3K4me3 in the *APX2* intergenic (R1) and genic (R2) regions, respectively, 3 h after heat regimen in Col-0 compared to the non-heat-treated Col-0 control. However, *csn5a-1* did not show any remarkable increment of H3K4me3 enrichment 3 h after heat stress. Three days after the heat regimen, the H3K4me3 levels returned to the baseline level in Col-0. For *csn5a-1,* the H3K4me3 levels increased by 3.13- and 3.03-fold 3 d after the heat regimen in the *APX2* R1 and R2 regions compared to the *csn5a-1* control (Figure 3A).

Next, we investigated H3K4me3 enrichment in *HSP22* and found a similar pattern. The H3K4me3 levels were 2.54- and 6.44-fold higher in Col-0; however, *csn5a-1* did not show any increase in the H3K4me3 levels 3 h following the heat regimen. Furthermore, the H3K4me3 levels returned to the baseline in Col-0 and increased by 3.09- and 2.5-fold in the R1 and R2 regions of *HSP22* in *csn5a-1* 3 d after the heat regimen (Figure 3B). When we investigated H3K4me3 enrichment in the non-memory gene *HSP70*, we again found a similar pattern. The H3K4me3 levels were 1.84- and 2.15-fold higher in the R1 and R2 regions of *HSP70* following 3 h of heat treatment in Col-0, which reached the baseline 3 d after heat stress. However, *csn5a-1* did not show any increase in the H3K4me3 levels 3 h after heat treatment but increased up to 3- and 2.8-fold in the R1 and R2 regions of *HSP70* 3 d after heat stress (Figure 3C). Taken together, these results show that the H3K4me3 levels increase in the intergenic (R1) as well as the genic (R2) regions of Col-0 3 h after heat stress in both memory (*APX2* and *HSP22*) and non-memory (*HSP70*) genes but it decreases to the baseline 3 d after heat regimen, while *csn5a-1* showed the opposite effect. In *csn5a-1,* enrichment of the H3K4me3 levels in the *APX2* and *HSP22* genes 3 d after the heat regimen show some correlation with the gene expression pattern of both genes; however, this enrichment is also found in the intergenic region of these genes, making it a general trend. Thus, a higher expression of *APX2* and *HSP22* 3 d after the heat regimen is not perfectly associated with higher levels of H3K4me3 in *csn5a-1*. Furthermore, the H3K4me3 levels were also high in *HSP70* of *csn5a-1* but not in Col-0 3 d after heat stress. However, the gene expression of *HSP70* was 3.5-fold higher in both *csn5a-1* and Col-0 3 d following heat treatment, suggesting that a change in gene expression levels following heat treatment is not associated with H3K4me3 methylation.

We observed no increase in H3K4me2 levels in the intergenic (R1) or in the genic regions (R2) of *APX2* 3 h after heat treatment in either Col-0 or in *csn5a-1*. However, 3 d after the heat regimen, the H3K4me2 level was elevated 1.48- and 1.5-fold in the intergenic (R1) and the genic (R2) regions of *APX2* of *csn5a-1* (Figure 4A), similar to H3K4me3 (Figure 3A). Similarly, no enhancement in H3K4me2 was detected in *HSP22* or *HSP70* in both Col-0 and *csn5a-1* 3 h following heat stress. However, the H3K4me2 levels were enhanced for both genes in *csn5a-1* 3 d after the heat regimen in *csn5a-1* (Figure 4B,C).

The difference in the enrichment of H3K4me3 from H3K4me2 is that the H3K4me3 (but not H3K4me2) levels increase in Col-0 3 h following heat stress in both the genic and intergenic regions of *APX2, HSP22,* and *HSP70*. However, both types of methylation are enriched in the R1 and R2 regions of *APX2, HSP22,* and *HSP70* 3 d after the heat regimen in *csn5a-1*. Thus, similar to H3K4me3, change in the gene expression level of memory genes following heat stress was also not correlated with H3K4me2 levels.

### 3.3. CSN5A Is Required for Deposition as Well as Restoration of the H3K4me3 Level 3 h and 3 d after Recurrent Heat Stress

Using the *APX2* genic region (R2) data, we prepared a model for H3K4me2 and H3K4me3 enrichment 3 h as well as 3 d following the heat treatment in Col-0 and *csn5a-1*. This model shows that the relative levels of H3K4me3 (but not H3K4me2) increases in heat-stressed Col-0 compared to the control Col-0 3 h after heat stress (Figure 5A). However, the H3K4me3 levels did not increase in *csn5a-1* 3 h following heat stress (Figure 5B). Interestingly, the H3K4me3 level returned to the baseline 3 d after heat stress in Col-0 (Figure 5A), whereas the levels of H3K4me3 and H3K4me2 increased in *csn5a-1* (Figure 5B) 3 d after heat treatment. The mutant (*csn5a-1*) shows defects in response to heat stress after 3 h as well as after 3 d. Thus, this model suggests that CSN5A is required for proper response to heat stress by increasing the levels of H3K4me3 3 h following the heat treatment, and more importantly, it is required to decrease the level of H3K4me3 to the baseline 3 d after heat stress (2 h, 44 °C, 7 d).

The gene expressions of the memory genes (*APX2* and *HSP22*) are not associated with the H3K4me3 levels; however, both the memory genes and H3K4me3 reached the baseline level 3 d following heat stress in Col-0. This suggests that CSN5A works independently to bring down the memory gene expression and the H3K4me3 level to the baseline 3 d after recurrent heat stress (2 h, 44 °C, 7 d).

## 4. Discussion

Transcriptional memory has been studied extensively in yeast and is an emerging topic in plants. Transcriptional memory is defined as a faster and higher induction of transcript levels after repeated stress following an intermediate lag with a concurrent maintenance of expression for a few days following stress [21,26,27]. There are already some reports on plants specifying the role of genes and transcription factors in maintaining transcriptional memory [20,22,28,29]. However, it is not known how the transcriptional memory is reset to its prestress state or which factors are responsible for resetting the transcription level to its original state. Here, using a viable hypomorphic mutant of the COP9 Signalosome, *csn5a-1*, we implicated CSN5A as a critical factor in resetting stress memory.

While the *csn5a-1* mutant seedlings responded as wild types to heat induction of the transcriptional stress memory genes *APX2* and *HSP22,* the inductions of *APX2* and *HSP22* were significantly greater in *csn5a-1* compared to Col-0. The elevated levels were maintained in *csn5a-1* three days following the heat regimen, while in the wild type, they reverted to baseline levels, indicating that stress-induced transcriptional memory remains longer in *csn5a-1*. This maintenance of heat-induced memory was specific for memory genes, as the expression of the non-memory gene, *HSP70,* did not show a similar pattern, indicating that CSN5A specifically influences transcript levels of the memory genes.

In global genome profiling, H3K4me3 was shown to correlate with the active transcription of genes [18], with H3K4me3 and H3K4me2 being enriched in heat and bacterial defense-primed genes [20,30]. Thus, we studied H3K4me3 and H3K4me2 to find a possible correlation of the transcriptional stress memory with histone methylation. Indeed, the H3K4me3 levels were many folds higher in the genic and intergenic regions of memory genes (*APX2* and *HSP22*) in Col-0 following the heat treatments. Surprisingly, the level of H3K4me3 was not elevated in *csn5a-1*, while we expected a higher-fold increment since the expressions of *APX2* and *HSP22* were several folds higher three hours following heat treatment. However, the H3K4me3 levels were increased nearly three-fold in both the genic and the intergenic regions of *APX2* and *HSP22* in *csn5a-1* 3 d following heat stress. Thus, the expression of memory genes is not perfectly correlated with the levels of H3K4me3.

This suggests that CSN5A regulates memory genes and H3K4me3 separately and that the enrichment of H3K4me3 in genes after recurrent stress is a general response independent of transcriptional activation. Our findings go hand in hand with a study showing that H3K4me3 does not correlate with transcriptional response at the single gene level after priming treatment [17]. Thus, an increase in the H3K4me3 level after stress is not related to a single gene, but it is a global phenomenon. Moreover, another study on rice showed that only 13% of genes were correlated with H3K4me3 levels following drought stress [31]. However, while Lamke et al., (2016) [20], showed a positive correlation between mRNA and H3K4me3 after heat stress, our heat regimens differed significantly. Thus, differences in our results are likely due to the variation in methodology of stress treatment, which can provide different results [17]. Furthermore, Howe et al., (2016) [32], showed that the majority of gene expressions were unaffected by the loss of H3K4me3, suggesting no positive correlation between gene expression and H3K4me3. Likewise, in our case, stress memory genes are not correlated with H3K4me3 because the expressions of *APX2* and *HSP22* were high in *csn5a-1* even though the H3K4me3 levels were low following 3 h of heat treatment. This further indicates a global instructive role of H3K4me3 in transcription.

Alternatively, it must be considered that, even though H3K4me3 and transcription are correlated, these can be independent events. For example, in mammals, the nucleosome-depletion region (NDR) with the appropriate base, such as a CpG island, recruits the histone lysine methyl transferase SET1A/B, leading to H3K4me3. Independently, NDR also recruits RNA polymerase II, leading to transcription. This shows that the correlation between H3K4methylation and transcription is not perfect; thus, it is feasible to obtain higher transcription levels without H3K4methylation and vice versa [32]. If the results suggest that H3K4me3 and H3K4me2 are not correlated with the initiations of transcription after heat stress, then what are the other functions to which histone methylation contributes following its elevation in wild types? It is suggested that H3K4me3 is not essential for transcription initiation and that it is instead deposited after transcription to regulate mRNA spicing, transcription termination, and transcriptional consistency [32]. Indeed, CSN is also suggested to have a role in RNA splicing [9].

In our study, we found that H3K4me3 methylation was many folds higher in *APX2* and *HSP22* of Col-0 3 h after heat treatment, but not in *csn5a-1*, suggesting that CSN5A is required for the deposition of H3K4me3 3 h after the heat treatment. Furthermore, CSN5A is required to reset the memory of H3K4me3 and the transcription expressions of *APX2* and *HSP22* to the baseline level 3 d after recurrent heat treatment, suggesting a role for CSN in resetting transcriptional stress memory and H3K4me3 methylation. Now, a question arises: does CSN regulate this transcriptional memory and histone methylation by regulating Cullin-based ubiquitin ligases or by some other mechanism? In correlation, the 19S proteosome is known to be involved in transcriptional regulation and H3K4me3 recruitment independent of its catalytic activity, suggesting that the 19S proteosome can regulate transcription regulation using different mechanisms. Considering that CSN is structurally and functionally similar to 19S proteosome [33], it may also regulate transcriptional memory and histone methylation in a deneddylation-independent manner.

This study adds to the work of Tuller et al. (2019) [9], who showed that the CSN regulates the epigenetic landscape of *Arabidopsis* to regulate transcription. Here, we show that CSN5 also regulates histone methylation and is crucial in resetting stress memory to a prestress state. This study opens up further questions such as whether the CSN directly interacts with histone methyl transferase to regulate the methylation level or if it is an indirect process interacting with other proteins and transcription factors.

## Figures and Tables

**Figure 1 biomolecules-11-00668-f001:**
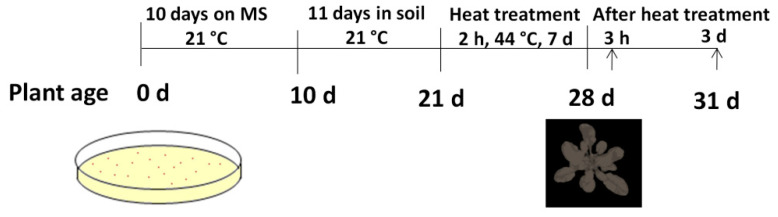
Schematic representation of experimental design; arrows denote the time of sampling.

**Figure 2 biomolecules-11-00668-f002:**
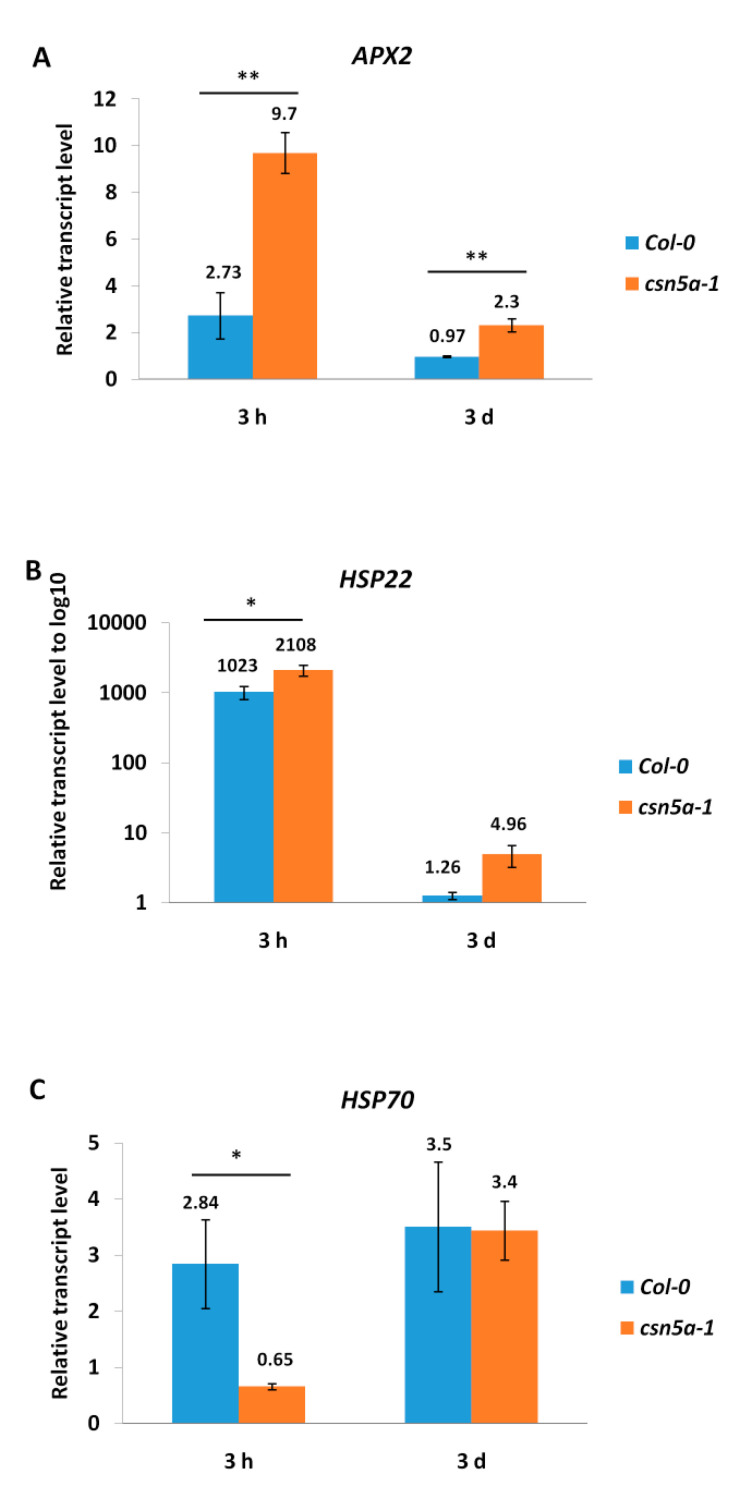
CSN5A regulates the heat stress memory genes after recurrent heat stress. (**A**) Relative transcript level of *APX2* 3 h and 3 d after recurrent heat stress (2 h, 44 °C, 7 d) in Col-0 (blue) and *csn5a-1* (orange). (**B**) Relative transcript level of *HSP22* 3 h and 3 d after recurrent heat stress (2 h, 44 °C, 7 d) in Col-0 (blue) and *csn5a-1* (orange). (**C**) Relative transcript level of *HSP70* 3 h and 3 d after recurrent heat stress (2 h, 44 °C, 7 d) in Col-0 (blue) and *csn5a-1* (orange). Transcript levels were normalized by *ACTIN* and the respective non-heat stress (control) sample at the same point. Note that the Y-axis is in log_10_ scale in graph (B). Error bars represent the standard error of the mean (SEM) of biological replicates (*n* = 3). Student’s t test * *p* < 0.05; ** *p* < 0.01.

**Figure 3 biomolecules-11-00668-f003:**
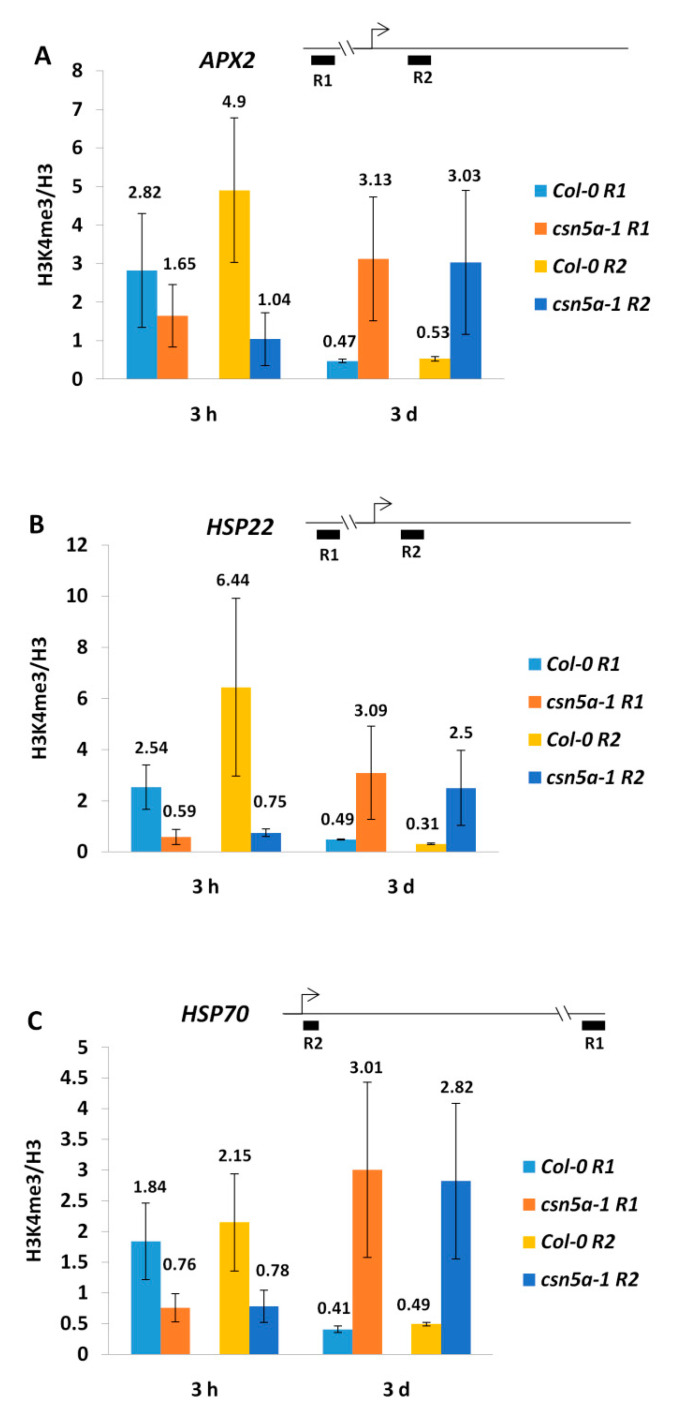
H3K4me3 fold change (heat stress/control) increases 3 h after recurrent heat stress in Col-0 and decreases to the baseline after 3 d; however, it is the opposite for *csn5a-1.* (**A**) H3K4me3 levels of *APX2* 3 h and 3 d after recurrent heat stress (2 h, 44 °C, 7 d) in Col-0 and *csn5a-1*. (**B**) H3K4me3 levels of *HSP22* 3 h and 3 d after recurrent heat stress (2 h, 44 °C, 7 d) in Col-0 and *csn5a-1*. (**C**) H3K4me3 levels of *HSP70* 3 h and 3 d after recurrent heat stress (2 h, 44 °C, 7 d) in Col-0 and *csn5a-1*. ChIP-qPCR was performed with antibodies against H3K4me3 and H3 (for normalization). Amplification values were normalized with H3 and the respective non-heat stress (control) sample at the same point. Schematics show the position of the gene region analyzed. R1 is the intergenic region, 3123 bp (*APX2*) or 2570 bp (*HSP22*) upstream of TSS, or 6725 bp (*HSP70*) downstream of the transcription start site (TSS). R2 is the 5′-region of *APX2, HSP22*, and *HSP70.* Error bars represent SEM of biological replicates (*n* = 3–4).

**Figure 4 biomolecules-11-00668-f004:**
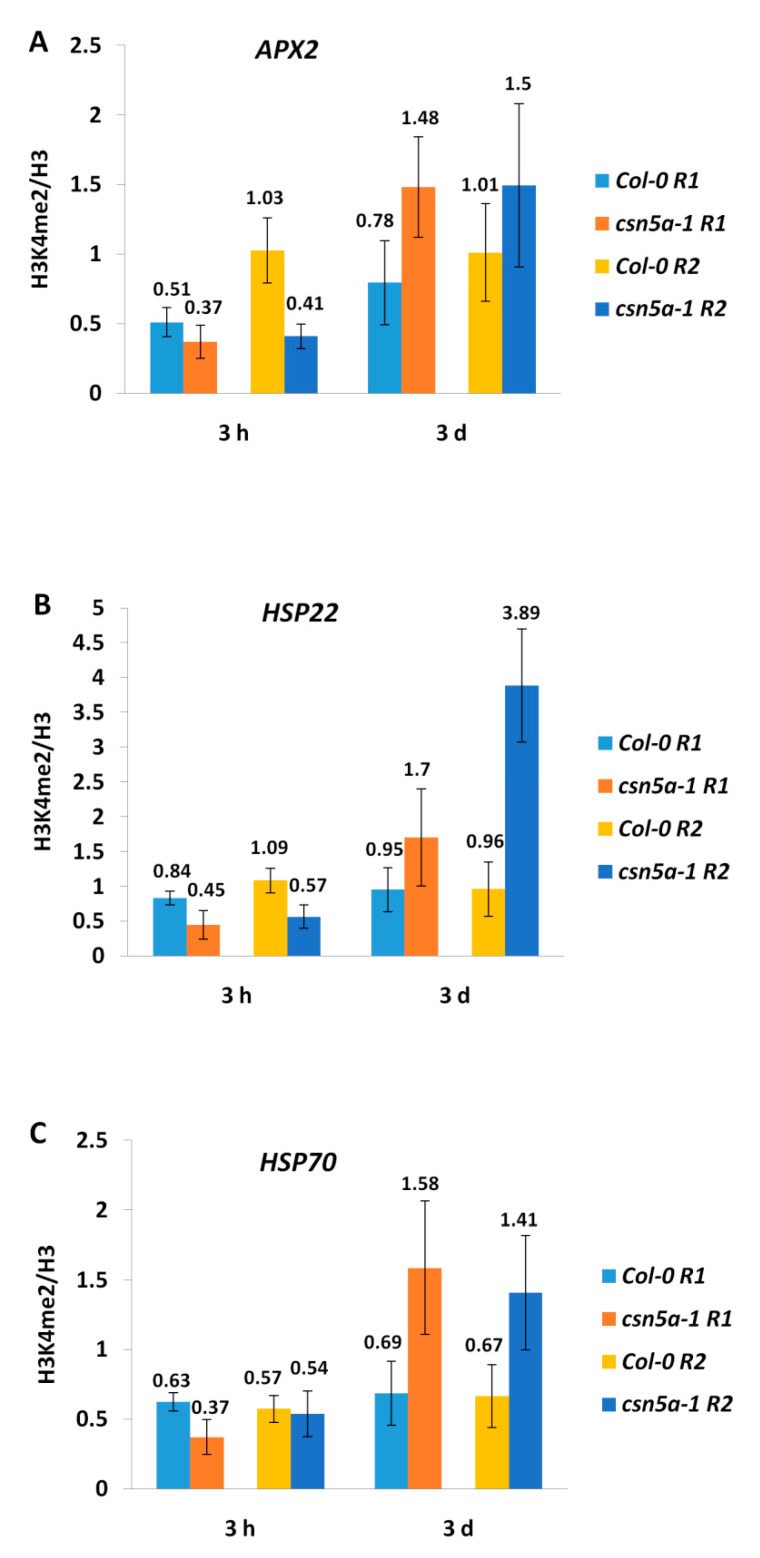
H3K4me2 fold change (heat stress/control) increases 3 d after recurrent heat stress in *csn5a-1*. (**A**) H3K4me2 levels of *APX2* 3 h and 3 d after recurrent heat stress (2 h, 44 °C, 7 d) in Col-0 and *csn5a-1*. (**B**) H3K4me2 levels of *HSP22* 3 h and 3 d after recurrent heat stress (2 h, 44 °C, 7 d) in Col-0 and *csn5a-1*. (**C**) H3K4me2 levels of *HSP70* 3 h and 3 d after recurrent heat stress (2 h, 44 °C, 7 d) in Col-0 and *csn5a-1*. ChIP-qPCR was performed with antibodies against H3K4me2 and H3 (for normalization). Amplification values were normalized with H3 and the respective non-heat stress (control) sample at the same point. Error bars represent SEM of biological replicates (*n* = 3–4).

**Figure 5 biomolecules-11-00668-f005:**
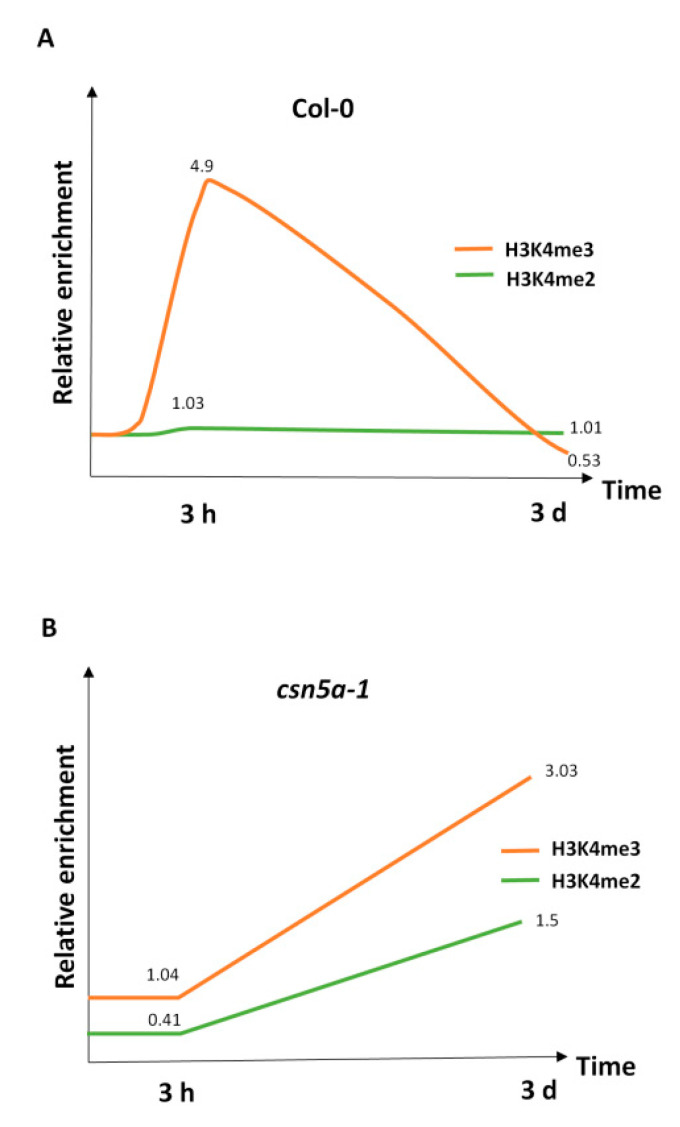
Restoration of H3K4me3 to the baseline 3 d after recurrent heat stress depends on CSN5A. (**A**) Presence of CSN5A induces the levels of H3K4me3 3 h after recurrent heat stress (2 h, 44 °C, 7 d); furthermore, CSN5A is also required to decrease the levels of H3K4me3 to the baseline after 3 d of recurrent heat stress. The H3K4me2 level is not affected by recurrent heat stress in Col-0. (**B**) In the absence of CSN5A (*csn5a-1*), the H3K4me3 level does not elevate 3 h of recurrent heat stress while levels of H3K4me3 as well as H3K4me2 elevate 3 d after recurrent heat stress. These figures were drawn based on *APX2* region 2 data.

## Data Availability

Not applicable.

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
