# Peer review of "CSN5A Subunit of COP9 Signalosome Is Required for Resetting Transcriptional Stress Memory after Recurrent Heat Stress in Arabidopsis"

_biomolecules, 2021, doi:10.3390/biom11050668_

Round 1

Reviewer 1 Report

In this report the authors further studied the role of CSN5 subunit of COP9 signalosome complex (CSN) and report alteration in the expression levels of stress memory genes APX2 and HSP22 and H3K4 methylation status in csn5a-1 mutant 3 hours and 3 days after application of heat stress. This work builds upon previous data by the authors and others, which showed that the growth of csn5a-1 was enhanced after heat stress. Under control condition the growth of csn5a-1 mutant is severely affected. This report provides additional evidence that csn5a-1 deletion introduces many abnormalities in plant indicating its pleiotropic effect. However few issues are to be clarified for a clearer view of how relevant are the results to advance in our understanding of CSN5 protein function in Arabidopsis.

Major comments

The experimental design is not clearly reported. The information relating to plant age, duration of heat stress and sampling time points is scattered at different subsections of Material and methods. I would recommend the authors provide a simplified scheme of the experimental design or more clear description.

Moreover, I would suggest the authors sow seeds directly into the soil to reduce the risk of seedling damage while transferring from MS plate to soil in future experiments. Inappropriate handling of seedlings adversely influences development of the plants and may modulate their response to stress conditions.

Did the authors verify how long was the expression of APX2 and HSP22 elevated after application of heat to csn5a-1 plants or how fast did it return to the baseline in wild-type plants? The results presented in this report showed that the expression levels of APX2 and HSP22 in csn5a-1 plants did not return to the baseline level at the same time as in wild-type plants. However it is not clear why was the expression of indicated genes analyzed only after 3 h and 3 days after treatment. Data from different time points: before, during and after the treatment for selected genes would provide more comprehensive insight into the response of csn5a-1 mutant to heat stress. 

Minor comments

Line 251: please replace the word “high” with “higher”

Line 280: the sentence is not clear, too general and poorly related to the presented results. It is contradictory to the results presented in Figure 1 and the earlier part of the Discussion section. Therefore I do not agree with the statement “…, similar to our case where expression of APX2 and HSP22 were unaffected in csn5a-1 even though H3K4me3 levels were low following heat treatment”.

Author Response

In this report the authors further studied the role of CSN5 subunit of COP9 signalosome complex (CSN) and report alteration in the expression levels of stress memory genes APX2 and HSP22 and H3K4 methylation status in csn5a-1 mutant 3 hours and 3 days after application of heat stress. This work builds upon previous data by the authors and others, which showed that the growth of csn5a-1 was enhanced after heat stress. Under control condition the growth of csn5a-1 mutant is severely affected. This report provides additional evidence that csn5a-1 deletion introduces many abnormalities in plant indicating its pleiotropic effect. However few issues are to be clarified for a clearer view of how relevant are the results to advance in our understanding of CSN5 protein function in Arabidopsis.

Major comments

Reviewer's comment 1: The experimental design is not clearly reported. The information relating to plant age, duration of heat stress and sampling time points is scattered at different subsections of Material and methods. I would recommend the authors provide a simplified scheme of the experimental design or more clear description.

Our response: As per the suggestion of the reviewer, we have added a schematic diagram as Figure 1 in the manuscript describing the experimental design and plant age. All the other figure numbers are changed accordingly in track change mode. 

Reviewer's comment 2: Moreover, I would suggest the authors sow seeds directly into the soil to reduce the risk of seedling damage while transferring from MS plate to soil in future experiments. Inappropriate handling of seedlings adversely influences development of the plants and may modulate their response to stress conditions.

Our response: Seeds are not sown directly in the soil because csn5a-1 mutant grows better when the seeds are sown in MS plate and transfer to the soil after 10 days, rather than directly sowing it on the soil. Special care was given while transferring the seedling from MS to the soil. Further, stress treatment was given at the age of 21 days which provides 11 day time for the plant to adapt to soil conditions and recover from any damage happened by chance while transferring from MS to the soil. 

Reviewer's comment 3: Did the authors verify how long was the expression of APX2 and HSP22 elevated after application of heat to csn5a-1 plants or how fast did it return to the baseline in wild-type plants? The results presented in this report showed that the expression levels of APX2 and HSP22 in csn5a-1 plants did not return to the baseline level at the same time as in wild-type plants. However it is not clear why was the expression of indicated genes analyzed only after 3 h and 3 days after treatment. Data from different time points: before, during and after the treatment for selected genes would provide more comprehensive insight into the response of csn5a-1 mutant to heat stress. 

Our response: The experimental design is similar to our previous work (Singh et al., 2019, Biomolecules), where we found that auxin signaling increases 3 h after heat treatment (2 h, 44 °C, 7 d) and it was maintained high even after 3 d heat treatment in csn5a-1 hypomorphic mutant, while it reverted to baseline level in wild type, indicating that csn5a-1 retains the memory of stress. Following the previous study, the aim of the present study is to check if transcriptional stress memory stays longer in the csn5a-1 hypomorphic mutant and if it stays longer; does it correlate with H3K4 methylation level? Present experimental design answers both the questions that transcriptional stress memory remains longer in csn5a-1 hypomorphic mutant compared to wild type and transcriptional stress memory is not perfectly correlated with H3K4 methylation. Hence, we did not study the gene expression level at any other time point.  

Minor comments

Reviewer's comment 1: Line 251: please replace the word “high” with “higher”

Our response: The word “high” has been replaced with “higher” now in line 261.

Reviewer's comment 2: Line 280: the sentence is not clear, too general and poorly related to the presented results. It is contradictory to the results presented in Figure 1 and the earlier part of the Discussion section. Therefore I do not agree with the statement “…, similar to our case where expression of APX2 and HSP22 were unaffected in csn5a-1 even though H3K4me3 levels were low following heat treatment”.

Our response: The sentence have been modified as “Howe et al., (2016) [32] showed that majority of gene expressions were unaffected by the loss of H3K4me3 suggesting no positive correlation between gene expression and H3K4me3, likewise in our case stress memory genes are not correlated with H3K4me3 because the expression of APX2 and HSP22 were high in csn5a-1 even though H3K4me3 levels were low following 3 h heat treatment” in line 290.

Reviewer 2 Report

This work of A.K. Singh and colleagues constitutes a new contribution to the incipient knowledge of how stress memory is set up in plants. Specifically, it reveals a role of CSN5A, a component of the COP9 signalosome, in transcriptional memory and histone methylation resetting after repeated heat stress episodes. The Arabidopsis loss of function mutant csn5a-1 shows a specific sustained expression of memory genes after the stress treatment, revealing that transcriptional memory remains longer in this mutant compared to the wild-type. This expression change it is not perfectly correlated with detected changes after the treatment in the H3K4me3 levels in genic and intergenic regions of the memory genes.

The research approach is similar to the one of Bäurle et al. in order to characterize the role of the transcription factor HSFA2 in heat stress memory (The EMBO Journal (2016) 35: 162–175). The paper is concise, according to a brief report, well written and illustrated. However, authors should address some minor points that will help readers the fully understanding of their work.

Lines 52 and 53 in page 2. Please, unify the criteria to refer to genes or proteins. I guess BRUSSHY1(BRU)/TONSOFU/MGOUN3, should not be in italics if you refer to the protein products of the gene.

Line 66. I would rather state “In a previous study” than “In the previous study”.

Line 76. Since one of the keywords of this article is “hypomorphic mutant”, I consider that it is necessary to present, even minimally, the information referring to the transgenic line csn5a-1, and not just refer it to a previous work.

Line 78. Here, and all over the text, there has been used the ordinal indicator instead of the degree symbol in temperatures figures. Please, check and correct.

Line 137 Correct “atleast”.

Figure 1. Three days after the treatment, is the expression difference of the memory gene HSP22 statistically significant or not? I guess it is, but I miss the line and asterisk(s) in that case.

Figure 1 legend. Line 207, please refer SEM for the first time in the paper as “standard error of the mean (SEM)”.

Figure 2 legend. Line 218, please refer TSS for the first time in the paper as “transcription start site (TSS)”.

Line 195. Please, consider rewriting of the first sentence of this paragraph. “Although the gene expression of memory genes (APX2 and HSP22) is not associated with H3K4me3 levels, both (memory genes and H3K4me3) reached to the baseline level 3 d following heat stress in Col-0.”

Line 247. Please correct “Signalsome”.

Author Response

This work of A.K. Singh and colleagues constitutes a new contribution to the incipient knowledge of how stress memory is set up in plants. Specifically, it reveals a role of CSN5A, a component of the COP9 signalosome, in transcriptional memory and histone methylation resetting after repeated heat stress episodes. The Arabidopsis loss of function mutant csn5a-1 shows a specific sustained expression of memory genes after the stress treatment, revealing that transcriptional memory remains longer in this mutant compared to the wild-type. This expression change it is not perfectly correlated with detected changes after the treatment in the H3K4me3 levels in genic and intergenic regions of the memory genes.

The research approach is similar to the one of Bäurle et al. in order to characterize the role of the transcription factor HSFA2 in heat stress memory (The EMBO Journal (2016) 35: 162–175). The paper is concise, according to a brief report, well written and illustrated. However, authors should address some minor points that will help readers the fully understanding of their work.

Reviewer's comment 1: Lines 52 and 53 in page 2. Please, unify the criteria to refer to genes or proteins. I guess BRUSSHY1(BRU)/TONSOFU/MGOUN3, should not be in italics if you refer to the protein products of the gene.

Our response: ItalicsBRUSSHY1(BRU)/TONSOFU/MGOUN3” has been changed to normal in line 62 and 63.

Reviewer's comment 2: Line 66. I would rather state “In a previous study” than “In the previous study”.

Our response: “In the previous study” has been replaced with “In a previous study” in line 66.

Reviewer's comment 3: Line 76. Since one of the keywords of this article is “hypomorphic mutant”, I consider that it is necessary to present, even minimally, the information referring to the transgenic line csn5a-1, and not just refer it to a previous work.

Our response: Following sentence has been added to describe minimally about the csn5a-1 in the materials and methods section “In brief, csn5a-1 show reduced growth with lower later root, deneddylation activity and auxin response” in line 76.

Reviewer's comment 4: Line 78. Here, and all over the text, there has been used the ordinal indicator instead of the degree symbol in temperatures figures. Please, check and correct.

Our response: Ordinal indicator has been replaced with degree symbol throughout the manuscript.

Reviewer's comment 5: Line 137 Correct “atleast”.

Our response: The word “atleast” has been replaced with “at least” in line 147.

Reviewer's comment 6: Figure 1. Three days after the treatment, is the expression difference of the memory gene HSP22 statistically significant or not? I guess it is, but I miss the line and asterisk(s) in that case.

Our response: We considered p < 0.05 for asterisk (*) however, p value for HSP22 is 0.09 so we have not added the asterisk and line in that case.

Reviewer's comment 7: Figure 1 legend. Line 207, please refer SEM for the first time in the paper as “standard error of the mean (SEM)”.

Our response: SEM has been referred as “standard error of the mean (SEM)” in Figure 1 legend now.

Round 2

Reviewer 1 Report

The authors have adequately addressed my concern.